# Extended Spectrum- and Carbapenemase-Based β-Lactam Resistance in the Arabian Peninsula—A Descriptive Review of Recent Years

**DOI:** 10.3390/antibiotics11101354

**Published:** 2022-10-05

**Authors:** John Philip Hays, Kazi Sarjana Safain, Mohammed Suliman Almogbel, Ihab Habib, Mushtaq Ahmad Khan

**Affiliations:** 1Department of Medical Microbiology and Infectious Diseases, Erasmus University Medical Centre (Erasmus MC), Rotterdam 3015 GD, the Netherlands; 2Department of Microbiological Sciences, North Dakota State University, Fargo, ND 58108, USA; 3College of Applied Medical Sciences, University of Hail, Hail 4030, Saudi Arabia; 4Department of Veterinary Medicine, College of Agriculture and Veterinary Medicine, United Arab Emirates University, Al Ain P.O. Box 1555, United Arab Emirates; 5Department of Medical Microbiology and Immunology, College of Medicine and Health Sciences, United Arab Emirates University (UAEU), Al Ain P.O. Box 15551, United Arab Emirates; 6Zayed Center for Health Sciences, United Arab Emirates University (UAEU), Al Ain P.O. Box 15551, United Arab Emirates

**Keywords:** antibiotic resistance, extended spectrum β-lactamase, carbapenemase, Arabian Peninsula

## Abstract

Antimicrobial resistance (AMR) is a global problem that also includes countries of the Arabian Peninsula. Of particular concern, is the continuing development of extended-spectrum β-lactamases (ESBLs) in the countries of this region. Additionally, antibiotic treatment options for ESBL-producing bacteria are becoming limited, primarily due to the continuing development of carbapenem resistance (CR), carbapenems being frequently used to treat such infections. An overview of recent publications (2018–2021) indicates the presence of ESBL and/or CR in patients and hospitals in most countries of the Arabian Peninsula, although the delay between microbial isolation and publication inevitably makes an accurate analysis of the current situation rather difficult. However, there appears to be greater emphasis on CR (including combined ESBL and CR) in recent publications. Furthermore, although publications from Saudi Arabia are the most prevalent, this may simply reflect the increased interest in ESBL and CR within the country. Enhanced ESBL/CR surveillance is recommended for all countries in the Arabian Peninsula.

## 1. Introduction

The bacterial extended-spectrum β-lactamase (ESBL) phenotype is mainly associated with members of the *Enterobacteriaceae* family, which are part of the normal gut microbiota. However, these bacteria may become pathogenic when associated with urinary tract and bloodstream infections [1]. Multiple enzymes and genes may confer the ESBL phenotype [2], thereby making treatment with β-lactam antibiotics, such as third-generation cephalosporins (e.g., cefotaxime, ceftriaxone, ceftazidime) problematic. Of particular concern is the rapid rise in the *bla_CTX-M-15_* enzyme, which was first detected in *Escherichia coli* isolated from India in 2001 [3] and has rapidly spread worldwide [4,5]. This type of AMR (i.e., CTX-M extended-spectrum β-lactamases) has been shown to have an environmental origin [6].

The consequence of the increasing resistance to cephalosporins has been a worldwide surge in the use of carbapenem antibiotics (originally effective against ESBL-positive bacteria). This phenomenon, in turn, is leading to the global and regional emergence and spread of carbapenem-resistant bacteria (mainly *Enterobacteriaceae*), potentially representing another formidable threat to public health [7,8,9,10]. Currently, carbapenem antibiotics, such as imipenem, ertapenem, meropenem, and doripenem, have a broad spectrum of activity and are crucial for treating life-threatening nosocomial (hospital-acquired) ESBL and MDR infections. However, the emergence of resistance to carbapenem antibiotics (mainly via the production of carbapenem-cleaving enzymes) is jeopardizing their use globally. The carbapenemase enzymes are categorized into four ‘Ambler’ classes based on their protein sequence similarity or their substrate and inhibitor profiles (Bush–Jacoby–Medeiros classification) [11]. The global spread of both ESBL and CR (sometimes carried in the same bacterial isolate) is a huge challenge to global healthcare systems, including those of countries in the Arabian Peninsula.

Countries in the Arabian Peninsula include Bahrain, Iraq, Jordan, Kuwait, Oman, Qatar, the Kingdom of Saudi Arabia (KSA), the United Arab Emirates (UAE), and Yemen [12]. Among the most common resistance phenotypes that have appeared in this region and globally within the past few decades are the ESBL and CR phenotypes. In this update, the authors highlight the most recent publications available in PubMed (https://pubmed.ncbi.nlm.nih.gov/) relating to ESBL and CR in the Arabian Peninsula between 2018 and 2021.

## 2. Results

The majority of recent publications in the Arabian Peninsula (from 2018 to 2021) relate to carbapenem-resistant bacteria. This emphasis is most likely due to the recent development of CR in this region. Additionally, Saudi Arabia had the most publications relating to ESBL and CR, which probably reflects differences in the amount of AMR research being performed in countries of the Arabian Peninsula rather than differences in AMR incidence.

### 2.1. Extended-Spectrum β-Lactamase Infections in the Arabian Peninsula 

JORDAN: Between June and October 2016, the analysis of 121 *E. coli* isolates from urinary tract infections among patients in two tertiary hospitals in Amman, Jordan, found that 62% (75/121) of the isolates were ESBL producers. *bla*_CTX-M-group-1_ genes were the most predominant (32/75), followed by combined *bla*_CTX-M-group-1_ and *bla*_SHV_ genes (15/75). Risk factors included previous hospitalization and use of a urinary catheter [13].

IRAQ: In Erbil, Iraqi Kurdistan, 81 *P. aeruginosa* isolates collected from hospitalized patients between 2012 and 2013 showed the presence of *bla*_VEB_ (30%), *bla*_prn_ (17%), and ESBL *bla*_PME_ (5%) [14]. In 2019, a study performed in the same city using urinary tract infection-associated isolates of *E. coli* (48 isolates) and *K. pneumoniae* (20 isolates) from 2016 indicated that the ESBL genes were distributed as follows: 81% *bla*_TEM_, 16.2% *bla*_SHV_, 32.4% *bla*_CTX-M_ and 64.7% *bla*_TEM_, 35.2% *bla*_SHV_, 41.1% *bla*_CTX-M_, respectively. The isolates were obtained from patients with thalassemia [15]. 

QATAR: Perez-Lopez et al. used whole-genome sequencing to determine the molecular characteristics of 254 *E. coli* and 73 *K. pneumoniae* isolates from children in Qatar, with the most frequent sequence types being ST131 and ST307, respectively. *bla*_CTX-M_-type ESBLs were found in 326 isolates, with the majority consisting of *bla*_CTX-M-15_ (87.8%) [16]. The presence of recently described colistin-resistance genes (*mcr-9* and *mcr-1*) has also been reported in ESBL-producing *K. oxytoca* and *E. coli* isolates from Qatar [17,18].

SAUDI ARABIA: A retrospective research of 72 (24 *E. coli* and 48 *K. pneumoniae*) suspected Neonatal Intensive Care Unit (NICU) outbreaks in Ha’il in 2014 showed that the majority of isolates (61 of *E. coli* and *K. pneumoniae*) were ESBL-positive, with high-level resistance to oxyimino-cephalosporins via the simultaneous production of *bla*_CTX-M-15_ and *bla*_SHV-12_. All *K. pneumoniae* isolates (42) were ST14, and all *E. coli* isolates (19) were ST131 [19]. Records from 2016 of a medical complex in Dammam (Eastern region of Saudi Arabia) showed that the ESBL isolates comprised 50% of the 176 *E. coli* and 42.1% of the 148 *K. pneumoniae* isolates [20]. Investigation of 211 clinical ESBL-producing *E. coli* isolates showed that 16 isolates were resistant to >10 antibiotics and the most common sequence type was ST131. The primary ESBL genes found were *bla*_CTX-M_ and *bla*_TEM_ [21]. From 2017 to 2018, 351 out of 1151 (30.5%) isolates obtained from clinical specimens at a University Medical college in Taif were found to be ESBL producers. These comprised 62.7% (220/351) *E. coli* and 23.6% (83/351) *K. pneumoniae*, with the distribution of the *bla*_CTX-M_ gene being higher (87%) than that of the *bla*_TEM_ (74.9%) and *bla*_SHV_ (29.4%) genes [22]. Sixty-seven of one hundred *E. coli* urine isolates from a hospital in Riyadh in 2018 had the ESBL phenotype; 31/33 (93.94%) of the isolates were associated with the *bla*_CTX-M_ gene, and all of them contained at least the *bla*_CTX-M-15_ gene [23]. A prospective, cross-sectional study of 311 Gram-negative bacteria (GNB) recovered from various clinical samples from a hospital in the Bisha Province between 2017 and 2018 indicated that 27% of the isolates were ESBL producers, most commonly of *E. coli* and *K. pneumoniae*. Interestingly, ESBL-producing GNB were most commonly found in the surgical and ICU departments, and co-production of ESBL and AmpC β-lactamases was found in 11.6% of GNB [24]. Diverse ESBL genes were found in MDR *Enterobacteriaceae* and *Acinetobacter baumannii* from clinical samples collected between 2017 and 2018. Seventy-eight out of one hundred seventy-three MDR isolates exhibited ESBL production in the presence of *bla*_TEM_ (84.7%), *bla*_CTX-M-15_ (33.3%), and *bla*_SHV_ (2.7%), as confirmed by PCR and sequencing [25]. The *bla*_CTX-M_ Group 1 and *bla*_CTX-M_ Group 26 genes were recently reported for the first time in 2018–2019 clinical isolates obtained from the Jizan region [26]. The presence of MDR (including ESBL- and carbapenem-resistant) hospital isolates of clonally related *K. pneumoniae* was reported in 2019 [27]. Alasmary reported that 8.94% (11/123) of uropathogens (mainly *E. coli* [58.5%] and *K. pneumoniae* [8.1%]) isolated in 2020 from three government hospitals showed ESBL resistance, stating that patients with UTIs in the Najran region are at a high risk of antibiotic resistance, which may be leading to significant problems in outpatient department treatment outcomes [28]. 

UAE: Ranjan Dash et al. reported that ESBL-producing *E. coli* and *K. pneumoniae* were responsible for 75% of community-acquired UTI in a hospital study between 2014 and 2016. The majority of these 399 ESBL isolates were also resistant to antibiotics (ciprofloxacin, 74% and trimethoprim-sulphamethoxazole, 73%) used to treat community-acquired infections [29]. In the same year, *bla*_CTX-M-28_ was reported in a setting where *bla*_CTX-M-15_ was predominant [30].

YEMEN: In 2013, 41 ciprofloxacin-resistant *E. coli* strains were isolated from a variety of clinical specimens, of which 63.4% (26/41) were ESBL producers. *bla*_CTX-M-15_ was detected in all the isolates. Additionally, 36.5% (15/41) of these isolates were genotyped as ST131 *E. coli* strains [31]. A study in two hospitals in Sana’a City, during 2014–2015, indicated that 44/130 ESBL-producing *E. coli* (33.8%) could be isolated from a range of clinical samples, including urine, wound exudates, sputum samples, blood, peritoneal fluid, vaginal swabs, and cerebrospinal fluid (CSF). These isolates were associated with the elderly, previously hospitalized patients with a hospital stay of more than 22 days, diabetic patients, and patients who had received a third-generation antibiotic therapy [32]. 

For a brief overview of the text see Table 1.

### 2.2. Carbapenemase-Based Infections in the Arabian Peninsula 

In a study of double-carbapenemase producers in the Arabian Peninsula, it was found that the location of the carbapenemase genes on transposons (Tn*2013*) and plasmids (IncHI1b, IncFIb/FII, and IncL/M) contributed equally to their emergence [33]. 

BAHRAIN: Of the 50 *P. aeruginosa* isolates from three hospitals in Bahrain (year of isolation not reported), 19 were found to carry *bla*_VIM_, one *bla*_NDM-1_, and one both *bla*_VIM_ and *bla*_NDM-1_ [34]. In another study, 549 of the 631 carbapenem-resistant *Enterobacteriaceae* (CRE) isolates collected within a 6-year period (2012–2017) were found to be *K. pneumoniae*, followed by *E. coli* (50/631). The elderly population and an ICU admission were found to be important risk factors for CRE acquisition. Notably, there was a rapid increase in CRE up to 2015 (45.4 cases per 10,000 patient admissions), followed by a gradual decrease in 2016–2017, which was attributed to the introduction of an ‘intense CRE control program’ involving infection control units and microbiology laboratories [35].

IRAQ: Using a PubMed literature search, Moghnieh et al. found 67 relevant articles from 2015 to 2020 related to AMR trends in Iraq, Jordan, and Lebanon and noted an increased CR in the *A. baumannii* species in all the three countries. Publications from Iraq and Jordan indicate that *Enterobacteriaceae* possess high rates of cephalosporin resistance. The most frequently observed resistance mechanisms in Gram-negative bacteria are genetic modifications, which facilitate the increased expression of antibiotic-inactivating enzymes and reduce bacterial cell permeability. Their conclusion was that there was a “concerning rise in AMR,” as well as a need for an improved understanding of the underlying mechanisms [36].

JORDAN: In Amman, 1% of clinical specimens (23/2759 isolates between 2013 and 2014) were found to have CR, including *K. pneumoniae*, *Enterobacter cloacae* complex, and *E. coli*, with the presence of *bla_NDM-1_*, *bla*_OXA-48_, and *bla*_VIM-4_ genes [37]. Subsequently, the analysis of 86 *A. baumannii* and *Acinetobacter* species isolates from three major hospitals in 2018 showed that 90.6% of them were carbapenem-resistant, with expression of the *bla_OXA-51-like_* (89.5%), *bla*_OXA-23-like_ (88.3%), and *bla*_NDM-1_ (10.4%) genes. The number of extensively drug-resistant (XDR) isolates was reportedly ‘alarming’ [38]. Hayajneh et al. used non-linear time-series methods to inform antibiotic policies to determine a threshold for third-generation cephalosporin and carbapenem use in a 533-bed tertiary teaching hospital in Irbid. Additionally, the use of alcohol-based hand rubs significantly reduced the incidence of carbapenem-resistant *A. baumannii* (CRAb), according to a retrospective study of ecological data collected between 2014 and 2019 [39]. In the same hospital, another retrospective study conducted between 2014 and 2019 concluded that antibiotic stewardship interventions led to significant reductions in antibiotic use (including third-generation cephalosporin and carbapenems) and had a significant impact on reducing the CRAb levels [40]. 

KUWAIT: The emergence of *bla*_OXA-181_ was found to be of concern in CRE from rectal swabs in two major hospitals between 2017 and 2018. Of the 590 patients examined, 58 were positive for CRE, with 38 colonized by *bla*_OXA-181_, 5 by *bla*_OXA-48_, and 1 by *bla*_KPC-2_ [41]. Asymptomatic intestinal carriage of carbapenem- and multidrug-resistant, ESBL-producing *Enterobacteriaceae* (mainly associated with *E. coli* and *K. pneumoniae*) was found to be relatively common in 405 food handlers (FH) in Kuwait between 2016 and 2018 [42]. Moghnia et al. also compared fecal CRE isolates between FH (n = 681) and patients in intensive care (ICP; n = 95) during the same period. Heterogeneous isolates were detected, with the distribution of *E. coli* in FH and ICP being 62.4% and 16.8%, respectively; the distribution was 2.6% and 72.6%, respectively, for *K. pneumoniae*. Thirty-six isolates were CRE (5.3%) in FHs, and 87 (91.6%) in ICPs, with a relatively high incidence of *bla_KPC_* in FHs and *bla*_OXA_ in ICPs [43]. 

More recently, the presence of β-lactamase genes, reduced bacterial permeability, and overexpression of MDR efflux pumps have been associated with 14 non-carbapenemase-resistant *Enterobacterales* in the same country [44].

OMAN: Whole-genome sequencing was performed on 35 putative carbapenem-resistant *E. coli* isolates recovered from patients at multiple centers in 2015. CRE (*E. coli* and *K. pneumoniae*) was found to be mediated by the production of *bla*_NDM_ and *bla*_OXA-48_ carbapenemases. Twenty-eight percent of these were *bla*_CTX-M-15_ producers [45].

QATAR: Qatar has also recently reported on CRE, focusing on the *mcr-1* (colistin resistance) gene in two *E. coli* isolates carrying *bla*_CTX-M-15_ and *bla*_NDM-1_, in pediatric ICU patients on routine screening [18,46]. Additionally, Pérez-López et al. investigated 72 carbapenemase-producing *Enterobacterales* from pediatric fecal specimen (61 children <18 years old) between 2018 and 2020 and found that *E. coli* (65.3%) and *K. pneumoniae* (30.6%) were the most common species. Genetic characterization of the 72 isolates indicated the presence of *bla*_NDM-5_ (30.8%), *bla*_NDM-1_ (19.2%), and *bla*_OXA-181_ (19.2%). The authors referred to the apparent sporadic introduction of antibiotic resistance within the healthcare setting in the Arabian Peninsula via asymptomatic carriers, particularly, those who visited or received healthcare in (nearby) countries with endemic antibiotic resistance [47]. In another study, 149 CRE (89 *K. pneumoniae* and 38 *E. coli*) isolates were found between 2014 and 2017, and the associated genes included *bla*_NDM-1_ (30.2%) and *bla*_OXA-48_ (19.5%) [48].

SAUDI ARABIA: Alotaibi reported on CRE during 2010–2018, indicating that 88% of the CRE isolates belonged to *K. pneumoniae*, and 11% to *E. coli* species [49]. In another study, *K. pneumoniae* and *Enterobacter cloacae* isolates from 2015 were found to contain *bla_NDM-1_*
_and_
*bla*_VIM-1_, respectively [27]_._ The carriage of these genes was found to be related to two large plasmids (>60kb), with isolates being MDR. Seventy-one *K. pneumoniae* isolates from Riyadh between 2011 and 2012 carried either *bla_OXA-48_* (67.6%) or *bla_NDM-1_* (8.5%), with *bla_CTX-M-15_* (66.2%) and *bla_CTX-M-14_* (21%) also being detected. Alsaleh et al. evaluated the appropriateness of carbapenem and piperacillin–tazobactam use in a tertiary-care hospital between 2016 and 2017. Of the 4929 prescriptions, only 2787 (58.6%) were considered to have been appropriately prescribed, with inappropriateness being attributed to too-broad a spectrum of activity (44.6%), antimicrobial use without culture (32.4%), and failure to suitably de-escalate the antimicrobials (19.9%) [50]. FIIK and L/M were the most predominant plasmid replicon types detected (69% and 67%, respectively) [51]. In Southern Saudi Arabia in 2015, a survey of 54 *K. pneumoniae* isolates with reduced sensitivity to carbapenem antibiotics indicated that the majority of isolates carried *bla_OXA-48_* (81.5%), and 7.4% carried *bla_NDM-1 (7.4%)_* [52]. Three years later, a collection of 519 carbapenem-resistant isolates from 13 Ministry of Health tertiary-care hospitals in five different regions of Saudi Arabia indicated that 84.7% (440/519) were CRE, with *K. pneumoniae* (456/591) being the most prevalent species. *bla_OXA-48,_ bla_NDM-1,_ bla_OXA-48_*, and *bla_NDM-1_* were the most prevalent carbapenemase genes detected [53]. A range of CR genes were found in 32 *A. baumannii* isolates from Taif in 2017, including *bla*_OXA-51_, *bla*_IPM_, *bla*_NDM_, and *bla*_OXA-23_ [54]. In addition, in 2017, *K. pneumoniae* isolates from the western region of Saudi Arabia were found to co-harbor the triple carbapenemase genes *bla*_KPC_, *bla*_NDM-1_, and *bla*_OXA-48_ [55]. In a selection of 200 carbapenem-resistant isolates (*K. pneumoniae*, *E. coli*, and *P. aeruginosa*) between 2014 and 2019, the most prevalent carbapenemase genes were *bla*_OXA-48_ (n=83, 41.5%), *bla_NDM-1_* (n=19, 2.5%), and both *bla*_OXA-48_ and *bla_NDM-1_* (n=5, 2.5%) [56]. In the Jizan province, 50 third-generation cephalosporin- and carbapenem-resistant isolates were collected between 2018 and 2019; there was a high incidence of *bla_CTX-M-_*_Group1_ cephalosporin resistance [26]. Worryingly, Al-Hamad et al. reported on clinical and environmental isolates from a hospital in the eastern region of Saudi Arabia in 2014, where 74 of the 208 (35.5%) surfaces tested were contaminated with carbapenem-resistant *A. baumannii*. Furthermore, all clinical and environmental isolates were found to be multidrug- or extremely drug-resistant. The range of carbapenemase genes found included *bla*_OXA-66_, *bla*_OXA-69_, ISA*ba*1-linked *bla*_OXA-23_, ISA*ba*1-linked *bla*_OXA-94_, *bla*_GES-1_, and *bla*_NDM-1_ [57]. High rates of MDR and *bla*_OXA-23_ and *bla*_OXA-51_ in *A. baumannii* were reported in isolates collected from 2006 to 2014 [58]. In 2018, the first report of carbapenem-resistant *Providencia stuartii* was published, although no genetic analysis was performed [59]. A *Salmonella enterica* serovar Kentucky isolate (first described in poultry) carrying *bla*_OXA-48_ was isolated from a Sudanese patient in Riyadh during a visit to Saudi Arabia in 2020 [60]. However, a preliminary search in PubMed indicated no further detection of this *bla*_OXA-48_-carrying strain in Saudi Arabia. Unsurprisingly, CR genes (in this case, *bla*_KPC_) have been found in a hospital sewage tank and municipal sewage treatment plant [61]. A retrospective, matched case–control study of pediatric patients with CRE infection in Riyadh, between January 2016 and 2017 indicated that recent surgery, antibiotic exposure, and the use of invasive procedures were significant risk factors for CRE colonization and infection [62]. 

UAE: Analysis of 295 carbapenem-producing *Enterobacteriaceae* from 2009 to 2014 indicated that 10.2% of them carried IncX3 plasmids, with *bla*_NDM-1_, *bla*_NDM-4_, *bla*_NDM-5_, *bla*_NDM-7_, *bla*_OXA-181_, or *bla*_KPC-2_ genes. The species represented were 13 *E. coli*, 13 *K. pneumoniae*, 2 *Enterobacter cloacae*, 1 *Citrobacter freundii*, and 1 *Morganella morganii* [63]. In a comparative study of two detection methods for carbapenem-resistant bacteria, a screening of 1813 rectal swabs from a clinic in Abu Dhabi over a period of 7 months yielded a positivity rate of 3.4%, mostly *K. pneumoniae* (22/61), with nine of these carrying the *bla*_OXA-48-like_ and *bla*_NDM-2_ genes [64]_._


YEMEN: Recently, Alsharapy et al. provided the first description of OXA-48-like-producing *Enterobacteriaceae* and NDM-5-producing *E. coli* in a hospital in Sana’a. Investigations into 27 isolates having decreased carbapenem-sensitivity (from a range of clinical specimens) showed that the majority of species were *K. pneumoniae* (18/27) and *E. coli* (6/27), with *bla*_NDM-1_ and *bla*_OXA-48_, and carrying the *bla*_OXA-232_ and *bla*_OXA-181_ genes [65]. 

Finally, although the situation may have changed in recent years, the majority (95.5%) of 1192 non-repeat CRE isolates from the Arabian Peninsula (as a whole) between 2009 and 2017 were found to be aztreonam–avibactam susceptible, followed by colistin- (79.8%), fosfomycin- (71.8%), and tigecycline- (59.9%) susceptible isolates [66]. In their study on FHs and ICPs, Moghnia et al. recommended that a multinational study be conducted to monitor the spread of AMR genes in the Arabian Peninsula [43]. In addition, the presence of *bla*_NDM-1_, *bla*_OXA-48_, *bla*_OXA-162_, *bla*_OXA-232_, *bla*_KPC-2_, and *bla*_NDM-1_ + *bla*_OXA-48_ was observed in 39 carbapenem-resistant *K. pneumoniae* ST14 strains isolated from 13 hospitals in Bahrain, Saudi Arabia, and the UAE between 2011 and 2016 [67].

For a brief overview of the text see Table 2.

## 3. Materials and Methods

A PubMed literature search was performed up to 23 December, 2021, using the following search criteria: ("extended spectrum beta lactamase"[Title] OR "extended-spectrum beta lactamase"[Title] OR "extended spectrum beta-lactamase"[Title] OR "extended-spectrum beta-lactamase"[Title] OR "extended spectrum β lactamase"[Title] OR "extended-spectrum β lactamase"[Title] OR "extended spectrum β-lactamase"[Title] OR "extended-spectrum β-lactamase"[Title] OR "ESBL"[Title] OR "ctx-m"[Title] OR "carbapenem"[Title] OR "carbapenemase"[Title]) AND ("Arabian peninsula"[Title] OR "bahrain"[Title] OR "iraq"[Title] OR "jordan"[Title] OR "kuwait"[Title] OR "oman"[Title] OR "qatar"[Title] OR "saudi arabia"[Title] OR "united arab emirates"[Title] OR "uae"[Title] OR "yemen"[Title]) AND ("2018/01/01"[Date - Publication]: "3000"[Date - Publication]). This generated 56 results, including one corrigendum. All publications were included in the current study and some publications (e.g., review articles) described research performed prior to 2018. Only English language articles were included (Figure 1), and where appropriate, risk factors associated with AMR were described. Searches of databases other than PubMed may have yielded different results.

## 4. Discussion

There is an inevitable lag between the initial isolation of microbial strains and the eventual publication of AMR results. Therefore, most of the studies published in 2018–2021 relate to isolates obtained several years before their publication. This makes an accurate analysis of the current state of affairs regarding the epidemiology of ESBL and CR in the Arabian Peninsula difficult, although this aspect of AMR epidemiological reporting is not restricted to this region of the world. However, recent publications have shown that there is an increasing interest in the epidemiology of CR in this region, most likely due to the increasing number of CR isolations from patients. Further, it is inevitable that the increasing use of carbapenem antibiotics to treat ESBL-related infections will lead to an eventual increase in CR globally, including in the Arabian Peninsula. As such, the steps required to help prevent or slow down the global emergence of CR infections include: i) recognition of the impact of carbapenem prescribing on regional infections and ii) ensuring that appropriate measures are taken towards the responsible prescription of antibiotics. In this respect, the need for a doctor’s prescription, a consultation with infection control teams, an accurate diagnosis of bacterial infections, and antibiotic sensitivity is essential [50]. It is encouraging to note that some of these steps are currently being implemented in the Arabian Peninsula [68,69,70,71]. 

From this review, it is obvious that most country-specific publications related to ESBL and CR infections were from Saudi Arabia. However, it should be remembered that this does not mean that Saudi Arabia has a larger AMR problem than other countries in the Arabian Peninsula, but suggests that the problem is potentially being more thoroughly investigated in this country [72]. 

Focusing on publications in PubMed may mean that a few relevant publications could have been omitted from this review. However, we described data obtained from 56 relevant publications, and it is unlikely that the impression obtained by this review would be altered by the addition of any additional publications that may have been missed.

## 5. Conclusion

Based on the current review, continued and expanded surveillance of ESBL and CR in hospitals and the environment is recommended for all countries in the Arabian Peninsula. Increased funding and the implementation of relevant AMR policies directed at non-specific and specific attributes associated with countries of the Arabian Peninsula, such as migration, religious pilgrimages, and food imports, are particularly recommended.

## Figures and Tables

**Figure 1 antibiotics-11-01354-f001:**
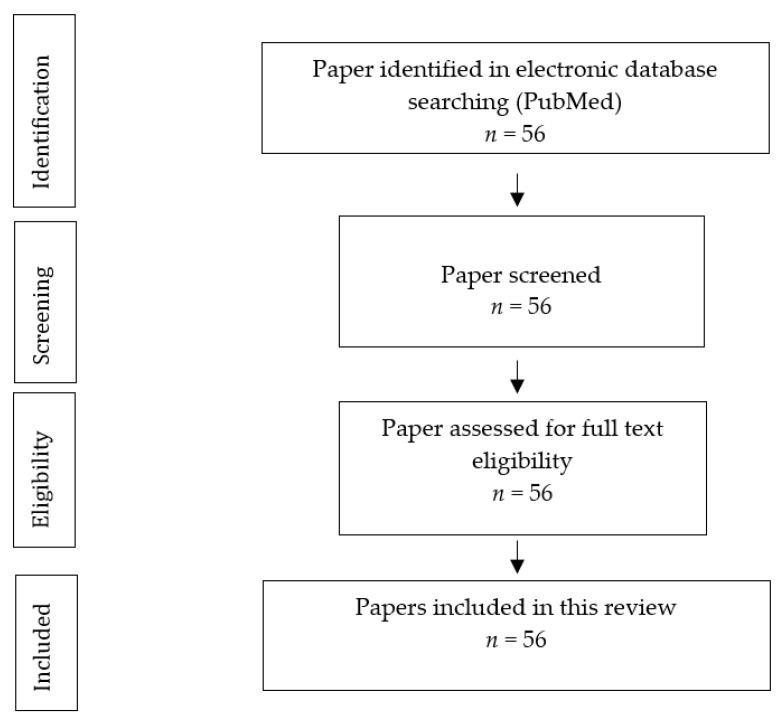
Flow chart summarizing study selection method.

**Table 1 antibiotics-11-01354-t001:** Investigations of Extended Spectrum β-Lactamase infections in the Arabian Peninsula.

Country	Sample	Organisms	Resistance Pattern	Resistance Genes	Reference
Jordan	Patients with UTI in two tertiary hospitals	*E. coli*	62% (75/121) were ESBL producers	Predominance of *bla*_CTX-M-group-1_ gene (42.7%), followed by combined *bla*_CTX-M-group-1_ and *bla*_SHV_ genes.	[13]
Iraq	Hospitalized patients	*P. aeruginosa*	-	*bla*_VEB_ (30%), *bla*_prn_ (17%), *bla*_PME_ (5%)	[14]
	Patients with UTI	*E. coli* and *K. pneumoniae*	-	81% *bla*_TEM_, 16.2% *bla*_SHV_, and 32.4% *bla*_CTX-M_ in *E. coli*. 64.7% *bla*_TEM_, 35.2% *bla*_SHV_, 41.1 % *bla*_CTX-M_ in *K. pneumoniae*	[15]
Qatar	Children	*E. coli* and *K. pneumoniae*		87.8% *bla*_CTX-M-15_	[16]
Saudi Arabia	Neonates in a suspected NICU outbreak	*E. coli* and *K. pneumoniae*	85% ESBL producers	100% *E. coli* harbored the CTX-M-15 gene. Among the *K. pneumoniae* isolates, 87.5% were ESBL positive with 92.85% contained the CTX-M-15 gene.	[19]
	Blood samples	*E. coli*	51.6% has ESBL phenotype	Predominance of *bla*_CTX-M-15_	[21]
	Clinical specimens	*E. coli* and *K. pneumoniae*	30.6% were ESBL producers	*bla*_CTX-M_ (87%), *bla*_TEM_ (74.9%) and *bla*_SHV_ (29.4%)	[22]
	Urine isolates	*E. coli*	67% had ESBL phenotype	93.94% associated with the *bla*_CTX-M_ gene	[23]
	Clinical specimens	*E. coli* and *K. pneumoniae*	27% were ESBL producers	Predominance of ESBL	[24]
	Clinical specimens	MDR *Enterobacteriaceae* and *Acinetobacter baumannii*	-	*bla*_TEM_ (84.7%), *bla*_CTX-M-15_ (33.3%) and *bla*_SHV_ (2.7%)	[25]
	Hospital isolates	*K. pneumoniae*	multidrug resistant (including ESBL and carbapenem resistance)	-	[27]
	Female patients with UTI	*E. coli* (58.5%) was the most commonly isolated organism, followed by *K. pneumoniae*(8.1%)	8.94% showed ESBL producers	8.94% showed ESBL resistance	[28]
United Arab Emirates	Community-acquired UTI	*E. coli* and *K. pneumoniae*	75% ESBL producers with resistance to ciprofloxacin (74%) and trimethoprim-sulphamethoxazole (73%)	-	[29]
Yemen	Clinical specimens	*E. coli*	63.4% (26/41) were ESBL producers	CTX-M-15 was present in 63.4% samples and qnrS in 12.2% samples.	[31]
	Elderly, previously hospitalized patients	*E. coli*	34% were ESBL producers		[32]

**Table 2 antibiotics-11-01354-t002:** Investigations of Carbapenemase-based infections in the Arabian Peninsula.

Country	Sample	Organisms	Resistance Pattern	Resistance Genes	Reference
Bahrain	Hospitalized patients	*P. aeruginosa*	-	38% *bla*_VIM_, 2% *bla*_NDM-1_, 2% *bla*_VIM_, 2% *bla*_NDM-1_	[34]
	-	*E. coli* and *K. pneumoniae*	87% were carbapenem-resistant *Enterobacteriaceae*	-	[35]
Jordan	Clinical specimens	*K. pneumoniae*, *Enterobacter cloacae* complex and *E. coli*	1% were carbapenem-resistant	*bla*_NDM-1_, *bla*_OXA-48_, and *bla*_VIM-4_	[37]
	Hospitalized patients	*A. baumannii* and *Acinetobacter* spp.	90.6% were carbapenem-resistant	*bla*_OXA-51_, (89.5%), *bla*_OXA-23_ (88.3%) and *bla*_NDM-1_ (10.4%)	[38]
Kuwait	Rectal swab samples	-	10% were carbapenem-resistant *Enterobacteriaceae*	*bla*_OXA-181_ and *bla*_OXA-48_	[41]
	FH and ICP	*E. coli* and *K. pneumoniae*	CRE- 5.3% FH isolates and 91.6% in ICP isolates	with a relatively high number of *bla*_KPC_ in FH isolates and *bla*_OXA_ in ICP isolates	[43]
Oman	Patient samples	Carbapenem-resistant *E. coli* isolates	-	59% carbapenemase-producing isolates were NDM and 23% were OXA-48	[45]
Qatar	Pediatric fecal carriers	Carbapenemase-producing *Enterobacterales*	-	*bla*_NDM-5_ (30.8%), *bla*_NDM-1_ (19.2%), and *bla*_OXA-181_ (19.2%)	[47]
	-	Carbapenem-resistant *E. coli* and *K. pneumoniae* isolates	-	*bla*_NDM-1_ (30.2%) and *bla*_OXA-48_ (19.5%)	[48]
Saudi Arabia	-	*E. coli* and *K. pneumoniae*	88% CRE	-	[49]
	-	*K. pneumoniae* and *Enterobacter cloacae*	-	*bla*_NDM-1_ and *bla*_VIM-1_	[27]
	-	*K. pneumoniae*	Carbapenem-resistant	*bla*_OXA-48_ (81.5%) and 7.4% carried *bla*_NDM-1_ (7.4%)	[52]
	-	Predominantly *K. pneumoniae*	84.7% were CRE	*bla*_OXA-48_, *bla*_NDM-1_, and combined *bla*_OXA-48_ and *bla*_NDM-1_	[53]
	Patient samples	*A. baumannii*	-	*bla*_OXA-51_, *bla*_IPM_, *bla*_NDM_ and *bla*_OXA-23_	[54]
	-	*K. pneumoniae*	-	*bla*_KPC_, *bla*_NDM-1_ and *bla*_OXA-48_	[55]
	-	*K. pneumoniae*, *E. coli* and *P. aeruginosa*	Carbapenem-resistant	*bla*_OXA-48_ (41.5%), *bla*_NDM-1_ (2.5%) and both *bla*_OXA-48_ + *bla*_NDM-1_ (2.5%)	[56]
	Clinical and environmental isolates	-	35.5% were carbapenem-resistant	-	[57]
		*A. baumannii*	-	*bla*_OXA-23_ and *bla*_OXA-51_	[58]
	Sudanese patient during a visit	*Salmonella enterica* serovar	-	*bla* _OXA-48_	[60]
	Sample from a hospital sewage tank	*-*	-	*bla* _KPC_	[61]
United Arab Emirates	-	*E. coli*, *K. pneumoniae*, *Enterobacter cloacae*, *Citrobacter freundii* and *Morganella morganii*	Carbapenem-resistant	*bla*_NDM-1_, *bla*_NDM-4_, *bla*_NDM-5_, *bla*_NDM-7_, *bla*_OXA-181_ or *bla*_KPC-2_ genes	[63]
	Rectal swabs	Predominantly *K. pneumoniae*	3.4% were carbapenem-resistant	40.9% carried *bla*_OXA-48_ and *bla*_NDM_	[64]
Yemen	-	Majority were *E. coli* and *K. pneumoniae*	Carbapenem-resistant	NDM-1, OXA-48, OXA-232 and OXA-181	[65]

## Data Availability

Not applicable

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
