# Peer review of "Extended Spectrum- and Carbapenemase-Based β-Lactam Resistance in the Arabian Peninsula—A Descriptive Review of Recent Years"

_antibiotics, 2022, doi:10.3390/antibiotics11101354_

Round 1

Reviewer 1 Report (Previous Reviewer 3)

The authors revised the article very professionally, and the revised article is more professional as compared to its initial draft. I recommend its publication in its current stage because the authors addressed all my questions very well. 

Author Response

No comments from the reviewer

Reviewer 2 Report (New Reviewer)

I will accept the descriptive review or narrative review as written.

Author Response

No comments from the reviewer.

Reviewer 3 Report (New Reviewer)

The authors study "Extended Spectrum- and Carbapenemase-Based β-lactam 3 Resistance in the Arabian Peninsula – A descriptive review" 

The study design is very poor. The number of studies and years chosen for this study is not sufficient. It should be 10 years or at least 5 years.  Therefore, this study provides no new information regarding antibiotic resistance and no clear snapshot of genetic determinants. The authors need to re-design this study and add data of last 10 years. The results section is very poor. The authors have just explained the results from 2010 to 2015 and they mixed the results and discussion part.

Some specific comments are suggested below;

Abstract:  The authors need to re-write the abstract as there is no results found from the current study.

Introduction

1. Line 53-56: Rephrase this sentence

2. Line 59-60: "This antimicrobial resistance (AMR) mechanism has been shown to have  an environmental origin"......What means by this sentence? Which AMR mechanism have an environmental origin ? confusing

3. Line 86: "Between 2018 and 2021" Why the authors just choose few years for such review study. As there are number of countries involved and only 3 to 4 years meta data will not be enough to give a full image of AMR from these countries.

Materials and Methods:

1. Line 102: - "Publication] : "3000"[Date - Publication])" Why choose only Pubmed and what 3000 means in this line ?

Results:

1. Line 119: "JORDAN - Between June and October 2016, 121 E. coli isolates" As this study was from 2018-2021, then why the authors discussing here the results of 2016 ???? 

Similarly same results were repeated like discussion. The authors have not properly discussed the results. Instead of results they wrote discussion part in the results section. While the discussion part in very poor.

The authors just extracted some resistance genes data. There is no Susceptibility data in this review. The authors need to show graphically the resistance profiles in these studies for every country which will be very interesting for readers. Also specify the prevalence data of different genes in different countries.

I am not able to write all the deficiencies of this review here. I am attaching  a PDF file for the authors.

Round 2

Reviewer 3 Report (New Reviewer)

The authors have not fully justified the comments in a better way.  I noted rather significant language, unsuitable choice of words, and sentence structure issues that need to be addressed. Please first send the manuscript to a MDPI English editor to improve the quality of English. It is not easy to review it in this form. However, I have some comments mentioned below need to be addressed after English Editing of the manuscript. Also check the attached file where I have highlighted.

Line 41: "If you mention CR "Carbapenem resistance" then no need to write AMR. Need to delete or rephrase"

Line 53-56: what the authors want to say ? Is this sentence is scientifically written ?

Line 59-60: The authors need to rephrase this whole sentence as it is not clear.

Line 59: Already mentioned in the abstract no need to repeat, just write AMR

Line 62: Why the authors write ESBL and AMR together ? instead of ESBL the authors need to mentioned cephalosporins ..... 

Line 61: This sentence should be rephrase, repetition of words should be remove

Line 81: "(and globally)" no need for brackets here

Line 82-83: Why the authors repeating the same words which are already mentioned above? No need to write the complete words if you have mentioned the abbreviations.

Line 89: MM_ Need to add one more section in methodology about the risk factors, selection criteria of data (better to mention that the samples mentioned in the publications were collected from when to when)

Line 103-104: What the authors means "all these publications were included in the current publication" ???? It seems you have not properly checked the previous comments and suggestions. Need extensive English editing with a professional english editor.

Line 110-111: rephrase the whole sentence

Line 122-123: write the values "followed by combined blaCTX - M -group - 1 123 and blaSHV genes."

Line 123-124: Risk factors should be mention in the MM section.

Line 137: better to write "with majority of blaCTX-M-15 (87.8%).

Line 137-138:  Colistin have many genes mcr (1-10). The authors need to mention the type of mcr 

Line 140: What is NICU. Need to write the full name

Line 141-142: Is the author means 61 were E. coli and 72 were K. pneumonia ?? Rephrase and correct the exact values

Line 171: multidrug resistance, The authors need to use abbreviation MDR throughout the manuscript

Line 178-181: Rephrase the sentence. Very poor English. The authors need to send the manuscript to a professional English Editor. Otherwise, I would not be able to review it again

Line 183-194: As just few lines above you use UTI as abbreviated now here again you mentioned both "full name and abbreviation" ??????

Line 205: "Tn2013" what is Tn2013???

Line 371: "Discussion" Discussion is too short and inappropriate. The authors need to revise it and add explanation

Line 384: is it meaningful sentence? Rephrase it

Author Response

This manuscript is a resubmission of an earlier submission. The following is a list of the peer review reports and author responses from that submission.

Round 1

Reviewer 1 Report

This is an important an detailed review on ESBL and CR in the Arabian Peninsula, reflecting the (current) prevalence of these transmissible resistances and closes an important gap in understanding the worlwide distribution of this AMR problematic.

Although the English reads fluent, I would very much welcome at least one Figure depicting or a Table, summarizing the results from the publications of the different countries. Currently this paper is rather tedious to read and would need a thourough make up in format. As an example, the Arabian Peninsula could be presented on a map, where the different resistance mechanisms are represented in pie charts or bar plots. The discussion is sound, but rather descriptive.  I would recommend to summarize the most prevalent ESBL type in the Peninsula and reformat. e.g. "our review shows that CTX-M-15 was the most prevalent ESBL gene mainly found on plasmid XYZ in E.coli etc etc, followed by XYZ, however XYZ"

minor comments:

- line 58: write out the word (e.g AMR) before you use abbreviations

Reviewer 2 Report

The manuscript authored by Hays et al. attempted to summarize the current situation of extended-spectrum- and carbapenemase-Based β-lactam resistance in the Arabian Peninsula.

The overall objective of this study is exciting and very important in antimicrobial resistance. However, the design and methodology of this study are not well developed and explained.

Notably, the search strategy used in this study is very narrow because the authors searched some keywords in the “title” of the database content. Extending the search to “title/abstract” with the MeSH term would bring more literature to the field. Also, the authors restricted their search to only a few years without explaining why? Furthermore, the database search was restricted to only PubMed. To better understand the systematic search strategy, the author can review PRISMA guidelines (PRISMA-S: an extension to the PRISMA Statement for Reporting Literature Searches in Systematic Reviews, PMID: 33499930).

Moreover, the authors did not mention exclusion and inclusion criteria, data extraction methods, quality assessment, and data analysis process. Therefore, it is evident that the authors did not try to minimize bias, which is essential in a systematic review. 

Reviewer 3 Report

The article entitled "Extended Spectrum- and Carbapenemase-Based β-lactam  Resistance in the Arabian Peninsula – A Systematic Review" is an interesting topic but the authors only used one search Engine PubChem where they only find 55 studies. 55 studies are nothing for review articles. The article is poorly organized. The authors used materials and methods, and the results and discussion section in their review articles? I never saw the Results and Discussion section ever in my life in any review article. A review article is already discussing other results then why need a discussion? There is no sub-headings, no tables, nothing. The conclusion has no correlation with the article.

  1. In the abstract, all the abbreviation needs to be explained for the first time used. e.g only ESBL is explained for the first time in the abstract but CR and AMR are not explained in the abstract. I don't know what is the meaning of Cr and AMR. so kindly follow this procedure for all article.
  2. For the studies why the tenure was only selected between 2018-and 2021? is there some specific reason?
  3. There are a lot of English mistakes, language error, and typo mistakes. Kindly check ur paper line by line from start till end. line 57: "detected in Escherichia 56 coli isolates from India". it should be "isolated" from india.
  4. The authors should mention a specific heading "literature search" not "materials and methods" after the introduction mentioning what kind of search engines were used to find papers? how the articles were screened? which kind of studies were selected and which were neglected. 
  5. why only PubChem was used? there are many search engines, as many of the studies are not available in PubChem.